# Improving Language Model Pretraining with Text Structure Information

## Abstract

Inter-sentence pretraining tasks learn from sentence relationships and facilitate high-level language understanding that cannot be directly learned in word-level pretraining tasks. However, we have found experimentally that existing inter-sentence methods for general-purpose language pretraining improve performance only at a relatively small scale but not at larger scales. For an alternative, we propose Text Structure Prediction (TSP), a more sophisticated inter-sentence task that uses text structure to provide more abundant self-supervised learning signals to pretraining models at larger scales. TSP classifies sentence pairs over six designed text structure relationships and it can be seen as an implicit form of learning high-level language understanding by identifying key concepts and relationships in texts. Experiments show that TSP provides improved performance on language understanding tasks for models at various scales. Our approach thus serves as an initial attempt to demonstrate that the exploitation of text structure can facilitate language understanding.

## 1 Introduction

General-purpose pretrained language models have been widely applied in natural language processing (NLP). The most representative model of these models is BERT (Devlin et al., 2019), which is pretrained simultaneously on two pretraining tasks: a masked language model (MLM) task, and a next sentence prediction (NSP) task. While MLM masks words and requires models to fill clozes, NSP is an inter-sentence task of predicting whether two texts are continuous. Inter-sentence tasks learn relationships between sentences and facilitate high-level language understanding that is not directly learned by word-level pretraining tasks (Devlin et al., 2019). However, the representative inter-sentence task, NSP, has been found to fail to improve performance (Liu et al., 2019; Yang et al., 2019). Although a few successors of NSP have been proposed, they are still not widely adopted and researched. In this paper, we show that the existing inter-sentence methods for general-purpose language pretraining are actually suboptimal; to improve on those methods' weaknesses, we then propose an alternative that redefines what is learned from sentence relations.

The existing general-purpose inter-sentence pretraining tasks include NSP (Devlin et al., 2019), which discriminates whether two texts come from different documents; sentence order prediction (SOP) (Lan et al., 2020), which discriminates whether two texts are swapped; and sentence structure objective (SSO) (Wang et al., 2020), which discriminates whether two texts come from different documents or are swapped. To investigate claims of improved performance, we experimented these methods at three different scales (*Small*, *Base*, and *Large*), which mainly followed the setting of BERT (Devlin et al., 2019). The model size and the amount of consumed data increased from *Small* to *Base* and then to *Large* scale (see Appendix A for the details). As seen in Figure 1, our experimental results show that the existing methods improved performance only at the *Small*-scale whereas they undermined or failed to improve performance for models at larger scales.

In investigating the reason for little improvement by the existing methods at larger scales, we noticed that all of the aforementioned methods split an input text into two segments and learn from only the relationship between the two segments, thus ignoring the text's underlying structure. As illustrated in Figure 2, text structure can be seen as the organization of information in texts. Without text structure, a text becomes a long continuous word sequence, which is hard to read and makes it difficult to identify key concepts and logical relationships for humans. This intuitive understand-

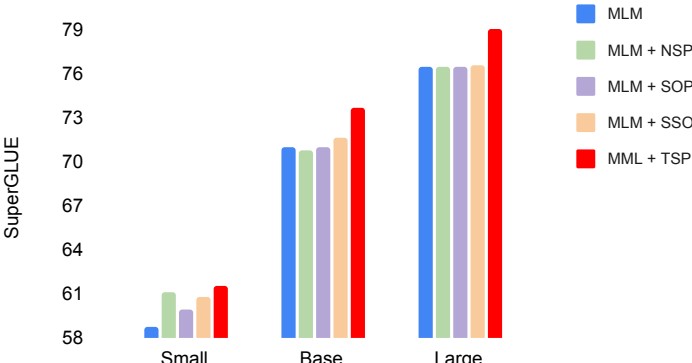

Figure 1: Experimental results on the language understanding benchmark SuperGLUE (Wang et al., 2019) at the *Small*, *Base*, and *Large* scales. The experiment compared the effect on the performance of using different inter-sentence tasks (NSP, SOP, SSO, and our proposed task, TSP) when learned concurrently with the word-based MLM pretraining task. By taking advantage of text structure information, TSP outperforms pure MLM at different scales, whereas the other inter-sentence baselines (NSP, SOP, and SSO) failed to improve the performance at larger scales in our experiments.

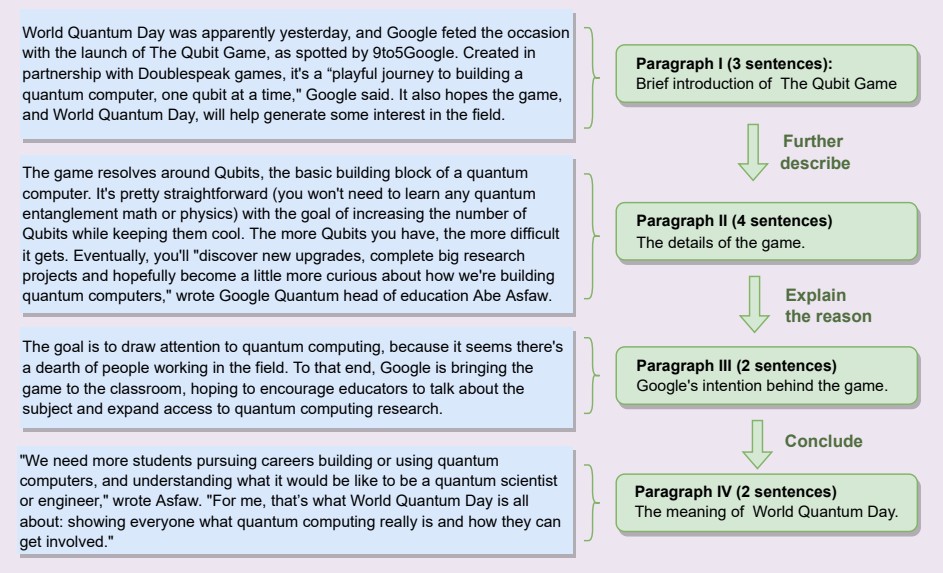

Figure 2: Example of text structure in real text. The document comprises multiple paragraphs, where each paragraph is composed of multiple sentences. Each paragraph conveys a specific concept constructed by the sentences that the paragraph is composed of, while the order of concepts comes with a clear intention and forms a logical flow. This suggests that the combination of hierarchy and order in text structure hides valuable clues to high-level language comprehension.

ing of human reading ability inspires us to explore the possibility that text structure can provide models' language understanding ability abundant learning signals of high-level semantics and their interaction, especially for models at larger scales. Hence, our goal in this paper is to demonstrate that learning from text structure can improve general-purpose language model pretraining. We thus propose a new inter-sentence task that better exploits text structure to examine our hypothesis.

The proposed task, Text Structure Prediction (TSP), is our initial attempt to integrate text structure information into general-purpose language pretraining. Regarding the task design, we view text structure in terms of two axes: ordering and hierarchy. Hierarchies are nested groupings of sentences, such as paragraphs or sections, where each group conveys a specific concept and its su-

perordinate group conveys a more global concept. The order of sentences represents the flow of context and connections between information such as chronological process or cause-effect relationship (Danes, 2015). Thus, the solution of the TSP task requires two abilities: understanding of the semantics of sentences to group them into nested concepts (reflected by hierarchical relationships), and identification of connective relations between them (reflected by ordering relationships). Specifically, we propose six inter-sentence relations for learning from text structure information, which combine three hierarchical levels and two ordering directions. As a result, the TSP task requires a model to classify sentence pairs over the six defined text structure relationships. More details of TSP are given in Section 3. Through experiments we have shown the following: (1) Our proposed method continues to improve performance on a language understanding benchmark at the *Small*, *Base*, and *Large* scales, whereas the inter-sentence baselines fail to do so. (2) Our proposed method is comparable to or better than the baselines on almost all language understanding tasks in the benchmarks. Accordingly, the results have shown the effectiveness and potential of exploiting text structure information for general-purpose language pretraining.

In summary, the contributions of this paper are as follows

1. We find that existing inter-sentence tasks for general-purpose language pretraining did not bring significant improvement in our settings. In our experiments, these approaches perform well at a relatively small scale, but fail to improve or even degrade performance at larger scales.

2. We propose the use of text structure information to provide more valuable learning signals for general-purpose language pretraining. For our initial attempt at this, we designed the Text Structure Prediction (TSP) task and show that TSP improves performance at all scales in our experiments. We thus demonstrate that the exploitation of text structure information is a promising research direction for facilitating general-purpose language understanding.

## 2 RELATED WORK

### 2.1 WORD-LEVEL PRETRAINING

Self-supervised word representation learning has rapidly developed in the past several years. GPT (Radford et al., 2018) predicts the next token with the information from the previous tokens. ELMo (Peters et al., 2018) uses a bidirectional framework to produce contextualized word representations by fusing the final hidden states of two LSTM networks processing information in different directions. BERT (Devlin et al., 2019) achieves deep bidirectional pretraining via the MLM task, which masks certain words in the input and then predicts the masked words. Since BERT's success, transfer learning has become mainstream in NLP. Many research initialize their models with pretrained models to reuse learned representations for their target tasks. Also, the development of word-level pretraining tasks has become prosperous. For example, ELECTRA (Clark et al., 2020) detects replaced tokens that come from incorrect predictions by another MLM model, while PMI-Masking (Levine et al., 2021) samples and masks collocations instead of words. In addition to encoder-based BERT and decoder-based GPT, word-level pretraining methods that are based on encoder-decoder such as T5 (Raffel et al., 2020) and BART (Lewis et al., 2020) are also developed, which gain great success on text generation tasks. Our proposed approach is learned simultaneously with a word-level pretraining task, and it can be easily combined with any word-level pretraining task that is based on transformer encoder, with negligible additional computational and memory costs.

### 2.2 SENTENCE-LEVEL PRETRAINING

Approaches for learning sentence representations mainly fall into three categories. The first type is **sentence bottleneck**, including Skip-Thought (Kiros et al., 2015) and CMLM (Yang et al., 2021), aims to generate single or multiple sentence representations that are useful for word prediction objectives. The sentence bottleneck methods compress the encoded sentence into bottleneck representations and thus benefiting from information bottleneck (Tishby et al., 1999). The second type is **contrastive learning**, including COCO-LM (Meng et al., 2021) and CLEAR (Wu et al., 2020), which makes representations of positive sentence pairs closer and vice versa. Contrastive learning methods improve uniformity and ease the anisotropy problem that limits the expressiveness of embeddings (Gao et al., 2021). The third type is **inter-sentence learning**, including NSP and SOP

described in Section 1, which learns sentence representations by capturing relationships between sentences. Since inter-sentence methods learn from interactions of sentences, they inherently also encode relationships between sentences and facilitate contextualization of multiple sentences. There is currently no research comparing the three kinds of methods with each other, and each type benefits from different principles. However, the inter-sentence approach is the only one that explicitly encodes and contextualizes multiple sentences. Our proposed approach belongs to the inter-sentence category.

### 2.3 INTER-SENTENCE TASKS

In terms of the kinds of sentence relationships to be learned, most inter-sentence tasks are sentence ordering tasks that rearrange a set of sentences into their original order (Cui et al., 2020; Kumar et al., 2020), whereas our proposed sentence relationships (Section 3.2) include ordering relationships but are not restricted to them. Regarding the creation of sentence representations, instead of encoding each sentence individually (Prabhumoye et al., 2020), our proposed framework (Section 3.1) adopts an approach of concatenating sentences into an input sequence, whereby information flows across sentences; thus, the created sentence representations are contextualized. In terms of semantic units, approaches such as NSP and SOP capture relationships between segments that are two halves of a text sequence that may include many sentences. In contrast, our approach processes sentences as in many other approaches. Finally, for task objectives, some approaches aim to identify coherent text (Barzilay & Lapata, 2008; Mesgar & Strube, 2018) or learning discourse representations (Iter et al., 2020), whereas our baselines (NSP, SOP, SSO) and our proposed method are for creating general-purpose language models.

## 3 PROPOSED METHOD

We explain our proposed method in two parts. Section 3.1 introduces the overall proposed framework for creating sentence embeddings and then performing six-class classification over sentence pairs using created sentence embeddings. Figure 3 shows an overview of this framework. Section 3.2 further elaborates on the six inter-sentence relationships that we designed for classification and learning over sentence pairs. Overall, the implementation of TSP comes with several advantages: (1) It is a self-supervised task that does not require external resources or human labeling. (2) It can easily be combined with any pretraining task that outputs contextual word representations. An example is provided in Appendix F. (3) It can be learned concurrently with other tasks in one forward pass. (4) It only requires constructing an output head on top of a text encoder, with negligible extra parameters and computations.

### 3.1 PROPOSED FRAMEWORK

#### 3.1.1 INPUT SEQUENCE

Given a text sequence, we mark the scope of each sentence in the sequence by using the NLTK toolkit (Bird et al., 2009), which is based on Punkt (Kiss & Strunk, 2006). The result is an sequence that can be described as $x = < x_{1,1}, x_{1,2}, ..., x_{1,n_1}, ..., x_{k,1}, x_{k,2}, ..., x_{k,n_k} >$, where $x_{i,t}$ is the $t$-th token in sentence $i$, $n_i$ is the length of sentence $i$, and $k$ is the number of sentences in $x$. Next, the sequence's sentences are shuffled to enable the model to learn to identify their correct order. We have found that shuffling all of a sequence's sentences introduces noise into the model; thus, we shuffle only a limited percentage of the sentences (15% in this work, as explained in Appendix C), while keeping the other sentences at their original positions. Additionally, to learn simultaneously from the MLM task, we follow the same procedure used in BERT to corrupt the sequence: 15% tokens are selected; then, 80% of them are replaced with a [MASK] token, 10% are replaced with random tokens, and 10% are unchanged. Finally, we add sentinel tokens *[CLS]* and *[SEP]* to the sequence as in BERT, and a masked and partially shuffled input sequence is thus obtained for the model.

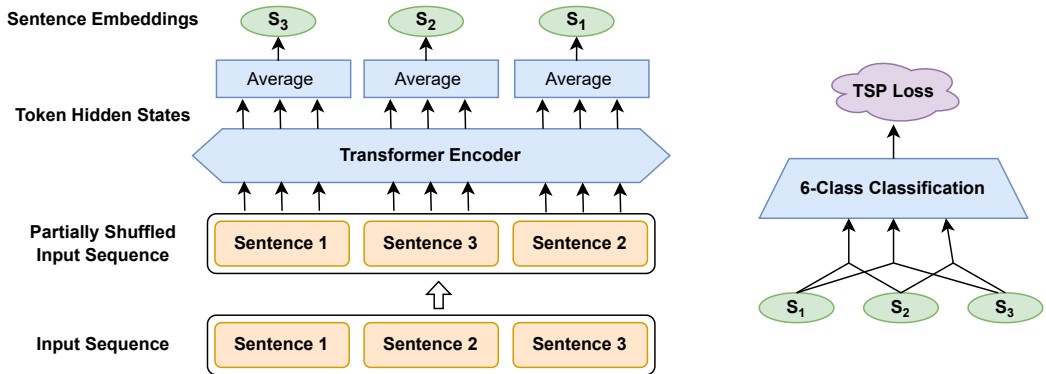

Figure 3: Overview of the proposed Text Structure Prediction (TSP) task. (left) Before an input sequence is sent to the encoder, certain randomly selected sentences in the sequence are shuffled. After encoding, a sentence embedding is obtained by averaging the encoded states of the tokens belonging to that sentence. (right) The TSP loss is calculated for performing six-class classification over sentence pairs by their sentence embeddings, where the six classes are shown in Figure 4.

### 3.1.2 SENTENCE EMBEDDING

To obtain sentence representations, we first calculate contextualized token hidden states:

$$h = f_\theta(x'), \tag{1}$$

where $f_\theta$ is the transformer encoder function and $x'$ is a corrupted and partially shuffled input token sequence. By averaging the hidden state vectors of tokens that belong to the sentence, we obtain its sentence embedding:

$$s_i = \frac{\Sigma_{t=1}^n h_{i,t}}{n_i}, \tag{2}$$

where $h_{i,t}$ is the hidden state vector of the $t$-th token in the $i$-th sentence and $n_i$ is the length of the $i$-th sentence. Additionally, through the transformer encoder's attention mechanism, each token interacts with all tokens in the input sequence rather than only the tokens in the sentence it belongs to. As a result, the sentence embeddings are contextualized and aware of other sentences in the same sequence. This means that when we compare sentences for their structure relationships, not only the sentences themselves but also their context are considered. Since identifying structure relationships using only information from the compared sentences can be very hard, it is intuitive that models are motivated to encode high-level interactions between sentences in the context into sentence representations.

### 3.1.3 PAIRWISE CLASSIFICATION

Next, we perform six-class classification over sentence pairs to learn the text structure relationship between sentences. Given a pair of sentences, we pass their sentence embeddings to a classifier to obtain prediction logits $z_{i,j} = W_2(GELU(W_1(s_i \oplus s_j)))$. $\oplus$ denotes vector concatenation, $s_k$ is the embedding of the $k$-th sentence, and $GELU$ is the activation function proposed by Hendrycks & Gimpel (2016). Finally, the loss in the Text Structure Prediction (TSP) task is the cross-entropy loss:

$$\mathcal{L}_{tsp}[i,j] = -log \frac{exp(z_{i,j,y_{i,j}})}{\Sigma_{c=1}^6 exp(z_{i,j,c})}, \tag{3}$$

where $y_{i,j}$ is the correct label for the relationship between the $i$-th and $j$-th sentences, and $z_{i,j,c}$ is the $c$-th element of the logit vector for the sentence pair. The final TSP loss $L_{tsp}$ is the average over the TSP losses for all sentence pairs in a mini-batch, where the sentences in each sentence pair are in the same sequence. Overall, a model is optimized by minimizing the sum of MLM loss $\mathcal{L}_{mlm}$ and our TSP loss $\mathcal{L}_{tsp}$:

$$\mathcal{L} = \mathcal{L}_{tsp} + \mathcal{L}_{mlm}. \tag{4}$$

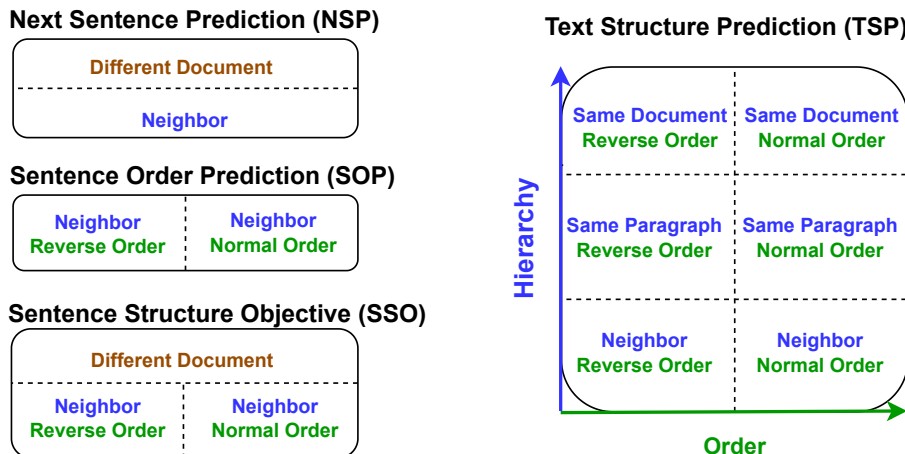

Figure 4: Inter-sentence relationship sets for (left) our baselines and (right) TSP. An inter-sentence task classifies pairs of texts into one of the relations in its own defined sets. For example, if two sentences are in the same paragraph (but not adjacent) and in reverse order, the TSP task should classify the pair as *Same Paragraph & Reverse Order*. Note that we leave *Different Document* for future work because of its lack of difficulty (Lan et al., 2020) and complexity of implementation. By comparing the defined relations, we can see that TSP takes better advantage of the text structure and is a more complex task that is expected to provide more valuable knowledge.

## 3.2 Proposed Text Structure Relationships

The key part of the training objective is our design of six classes to represent six text structure relationships between sentences. As illustrated on the right in Figure 4, these relationships are the set of combination of three hierarchical relationships {*neighbor, same paragraph, same document*} and two ordering relationships {*normal order, reverse order*}. Concretely, these relationships are defined as follows. The hierarchical relationships include (1) *neighbor* (two sentences are adjacent), (2) *same paragraph* (two sentences are in the same paragraph but not adjacent), and (3) *same document* (two sentences are in the same document but not adjacent or in the same paragraph). The ordering relationships include (1) *normal order* (two sentences are in the forward order), and (2) *reversed order* (two sentences are in the reversed order).

## 3.3 Learning Expectation from Hierarchy and Order

The text hierarchy describes the semantics hierarchy and the interactions between the semantics. As shown in Figure 2, the semantics of sentences are included in higher-level paragraph semantics, and paragraph semantics are included in higher-level document semantics. Relationships between semantics at different levels and the same level can also be seen in the example. To group sentences nestedly, models are expected to summarize semantics at different levels from the context, while capturing relationships and interactions between these semantics. Intuitively, it can provide unique learning signals for models' understanding ability on and between texts at different levels of granularity.

The order of sentences depends on the logical relations that connect the different concepts described in the sentences. These relations can be chronological relations, cause-effect relations, and so on. Textual coherence is also an important clue for sentence order. A model must recognize relations between concepts and identify coherence to rearrange the order of sentences. Moreover, learning from sentence order has long been shown to be useful for many downstream applications (Cui et al., 2020).

## 4 EXPERIMENTS

### 4.1 SETUP

#### 4.1.1 PRETRAINING

All models in this paper were pretrained on OpenWebText (Gokaslan & Cohen, 2019), an open-source recreation of the WebText corpus described in GPT2 (Radford et al., 2019). It includes 38GB of texts from 8,013,769 documents extracted from Reddit posts. Note that we did not use Wikipedia+BookCorpus because of the public unavailability of official BookCorpus (Zhu et al., 2015). In preprocessing the pretraining data, we delimited paragraphs by blank lines and marked the scopes of sentences with the sentence splitter of NLTK (Bird et al., 2009) for the use of TSP.

For this paper, we experimented with pretraining models on word-based MLM with different inter-sentence tasks or on MLM only. In particular, we mainly followed BERT's pretraining setting and referred to the setting of ELECTRA (Clark et al., 2020), another popular pre-trained model. We trained with a sequence length of 512 and 1 million training steps at the *Base* and *Large* scales, or with a length of 128 and 1.45 million steps at the *Small* scale. The batch sizes vary with the scale, as described along with other details in Appendix A. To ensure fair comparisons, all models at the same scale had almost the same computational cost and number of parameters in both training and inference. Note that we did not perform any hyperparameter search, although we think it could help our proposed method to perform better.

#### 4.1.2 EVALUATION

Our pretrained models were evaluated on the widely adopted SuperGLUE benchmark (Wang et al., 2019) for evaluating general language understanding systems. SuperGLUE contains tasks covering natural language inference tasks RTE (Dagan et al., 2006) and CB, multiple-choice reasoning task COPA (Melissa et al., 2011), question-answering tasks BoolQ (Clark et al., 2019), MultiRC (Khashabi et al., 2018), ReCoRD (Zhang et al., 2018), and word sense disambiguation task WiC (Pilehvar & Camacho-Collados, 2019). The metrics used here for these tasks are the same as those the ones described in the SuperGLUE paper.

Because some of the evaluation datasets are small and the scores on them could vary substantially depending on the random seed, we followed the ELECTRA setting and finetuned 10 runs for each task from the same pretrained checkpoint for every model. Here, we took the median score of 10 runs for our development set results (Appendix D), and we used the best-performing model on the development set for evaluation on the test set. In contrast to certain related papers, we did not apply tricks such as intermediate finetuning or ensembling during the stage of finetuning. Similarly, we excluded the WSC task because it involves intermediate finetuning and publicly unavailable training data to obtain decent results (Kocijan et al., 2019). More details about our finetuning are given in Appendix B.

### 4.2 RESULTS

Table 1 lists the averaged SuperGLUE scores. TSP improved on pure MLM and outperformed the inter-sentence baselines at all scales. These results show the effectiveness and potential of incorporating text structure information into learning. The results also reveal two findings. First, NSP undermined or made little difference in the performance, which matches the claims of Liu et al. (2019); Yang et al. (2019). Second, while SOP has been claimed to improve NSP on the GLUE benchmark (Wang et al., 2018; Lan et al., 2020), we found that it failed to outperform NSP in our experiment.

Next, we analyze the scaling of the four inter-sentence tasks. Although the inter-sentence baselines (NSP, SOP, and SSO) improved on pure MLM at the *Small* scale, at the larger scales they failed to improve on pure MLM by a significant margin or even undermined the performance. We suspect the reason to be the task difficulty. While the *Small*-scale models learned from the inter-sentence baseline tasks, those tasks may not have been difficult enough to provide valuable learning signals to models at larger scales. We can also see that SSO (the fusion of NSP and SOP) is slightly more complicated than NSP and SOP, and it slightly outperformed both NSP and SOP at *Base*

Table 1: Experiment results on the SuperGLUE test set at different scales. Overall, TSP outperformed MLM at the different scales, whereas the other inter-sentence baselines (NSP, SOP, and SSO) failed to achieve improvement. Note that because the sizes and settings for the naming of scales (e.g., *Small*, *Base*, and *Large*) differ greatly in different papers, results may not be directly comparable between papers.

| Scale | Model | Avg. | CB | COPA | MultiRC | RTE | WIC | BoolQ | ReCoRD |
|---|---|---|---|---|---|---|---|---|---|
| | | | Avg. F1/Acc. | Acc. | F1a/EM | Acc. | Acc. | Acc. | F1/Acc. |
| Small | MLM | 58.8 | 75.7/81.6 | 57.6 | 60.3/13.6 | 58.3 | 60.9 | 70.4 | 49.4/48.6 |
| | MLM+NSP (BERT) | 61.1 | 76.2/82.0 | 58.4 | 61.1/**14.7** | 65.5 | 65.5 | 71.5 | 50.1/49.3 |
| | MLM+SOP | 59.9 | 77.7/80.4 | 59.0 | 61.7/14.5 | 63.0 | 60.4 | 71.7 | 48.3/47.4 |
| | MLM+SSO | 60.8 | 74.4/81.6 | 57.2 | 61.1/14.3 | **66.1** | **66.1** | **71.8** | 49.3/48.6 |
| | MLM+TSP (ours) | **61.5** | **83.0/86.0** | 59.4 | 62.4/14.3 | **66.1** | 60.2 | **71.8** | **50.5/49.8** |
| Base | MLM | 71.0 | **84.8**/90.4 | 63.2 | 73.2/28.6 | 71.3 | 66.1 | 79.4 | 79.1/78.2 |
| | MLM+NSP (BERT) | 70.8 | 84.5/90.8 | 66.0 | 73.8/30.6 | 73.2 | **68.8** | 70.9 | 77.9/76.9 |
| | MLM+SOP | 71.0 | 83.5/90.4 | 70.0 | 71.7/25.4 | 70.8 | 68.1 | 78.3 | 74.9/74.0 |
| | MLM+SSO | 71.6 | **84.8**/89.6 | 71.2 | 73.0/28.5 | 72.3 | 68.0 | 76.9 | 75.0/74.0 |
| | MLM+TSP (ours) | **73.6** | 84.2/**91.2** | **72.4** | **75.8/33.5** | **74.1** | 66.6 | **80.6** | **79.2/78.5** |
| Large | MLM | 76.5 | 87.4/92.4 | 73.6 | 77.8/37.5 | 76.6 | 68.7 | **82.7** | **86.5/85.7** |
| | MLM+NSP (BERT) | 76.5 | 83.6/89.6 | 76.0 | 78.2/38.4 | 77.4 | 71.0 | 81.7 | 85.1/84.4 |
| | MLM+SOP | 76.4 | 84.8/90.4 | 74.6 | 79.3/40.9 | 77.1 | 68.6 | 82.2 | 85.2/84.5 |
| | MLM+SSO | 76.6 | 81.8/88.4 | 74.6 | 79.6/40.9 | 78.5 | 70.6 | 82.1 | 85.1/84.3 |
| | MLM+TSP (ours) | **79.0** | **93.5/94.8** | **77.4** | **80.5/43.2** | **79.1** | **72.4** | **82.7** | 85.8/85.2 |

and *Large* scales. On the other hand, with the better use of text structure information, TSP was designed as more complicated (six inter-sentence relations instead of two or three inter-sentence relations for the inter-sentence baseline tasks, as shown in Figure 4) and more difficult. TSP provides valuable knowledge for models at all scales, especially for *Large*-scale models, which have the largest capability and consumed data to better exploit the difficult TSP task, thus yielding the largest improvement over other baselines as compared to the other scales. Encouraged by this observation, we expect that models with even larger model sizes and training data would also benefit significantly from learning text structure information.

While TSP provided competitive or better performance on most of the tasks, we also found that it performed well on the reasoning (CB, RTE, and COPA) and question-answering (MultiRC and BoolQ) tasks. We can view a question and an answer as two texts that share a similar concept and can be placed in the same hierarchical group, such as a paragraph or a pair of consecutive sentences. Likewise, we can view reasoning in terms of whether two texts are coherent. From these viewpoints, the results are consistent with our interpretation that TSP facilitates language understanding by strengthening recognition and comparison of concepts and reasoning relationships (Section 3.2). On the other hand, although ReCoRD is a QA task, it is more like prediction of a masked entity in a paragraph given the context, which may explain why pure MLM was the leading model for ReCoRD at the *Large* scale. Regarding WiC, a word sense disambiguation task, TSP showed an interesting phenomenon. Compared to other baselines in the WiC task, TSP performed poorly at the *Small* scale, got better at the *Base* scale, and became the best-performing model at the *Large* scale. We can relate this result to our surmise on the scaling effect mentioned above. Learning of the complicated TSP stretches models' capabilities, and small models may not have enough capability to encode information that is useful for word-level tasks like WiC, whereas large models have enough capability that complicated sentence-level learning can even help with word-level understanding. While more elaborate approaches will be needed to analyze the effect of learning text structure information on different types of downstream tasks, which is outside the scope of this paper, we hope to have a good starting point by providing interesting results and explanations for those who follow.

In addition to the main results, we also reported other experiment results. First, we reported scores on the SuperGLUE development set in Appendix D, which showed similar results as in Table 1. Second, we reported results on the GLUE benchmark in Appendix E. Finally, to demonstrate that our proposed approach is generally useful, we did an experiment showing that TSP can be easily combined with another word-based pretraining task (see Appendix F for details). Additionally, this

Table 2: Ablation results for the designed text structure relations.

| Model | #total classes | #ordering relationship | #hierarchical relationship | SuperGLUE |
|---|---|---|---|---|
| MLM+TSP | 6 | 2 | 3 | 73.9 |
| MLM+TSP - paragraph | 4 | 2 | 2 | 73.2 |
| MLM+TSP - hierarchy | 2 | 2 | - | 72.5 |
| MLM+TSP - order | 3 | - | 3 | 71.9 |
| MLM | - | - | - | 71.2 |

experiment also shows that TSP still improves performance in its preliminary results, which suggests that learning text structure information may generalize to pretraining tasks other than MLM.

## 5 ABLATION STUDY

To investigate whether the proposed TSP text structure relations are helpful, we experimented with different TSP variations in which some text structure relations were ablated. Specifically, *MLM+TSP -hierarchy* adopted only the 2 ordering relationships (normal, reversed), *MLM+TSP - order* adopted only the 3 hierarchical relationships (neighbor, paragraph, document), and *MLM+TSP -paragraph* combined the 2 ordering relationships with only the 2 hierarchical relationships (neighbor, document) , with no discrimination of whether two sentences were in the same paragraph.

The results on the SuperGLUE development set of models at the *Base* scale are shown in Table 2. First, we found that the models pretrained with TSP benefitted from both the hierarchy relationships and the ordering relationships, as both *MLM+TSP - hierarchy* and *MLM+TSP - order* underperformed *MLM+TSP* and outperformed *MLM*. Second, the performance slightly dropped when it was not discriminated whether two sentences were in the same paragraph, which shows that the adjacency of sentences is not the only important factor, and that discrimination of hierarchical groups at different levels matters. Although other design elements such as discrimination of sections and documents remain unexplored, we will encourage future work to explore further, whereas in this paper we focus on uncovering the potential for integrating text structure information.

## 6 CONCLUSION

Despite their stated usefulness for high-level language understanding and continued development, inter-sentence tasks have not gained enough attention as expected. Our experiments have shown that existing inter-sentence approaches for general-purpose language pretraining can not improve performance or even degrade performance at scales larger than *Small*. Furthermore, these existing approaches ignore a large part of text structure information and oversimplify complicated text structures into one relationship between two lengthy parts of a text sequence. The goal of this paper was thus to explore the potential of utilizing text structure information to provide more valuable learning signals and maintain the improvement at larger scales. For an initial attempt to achieve this goal, we proposed Text Structure Prediction (TSP), a self-supervised inter-sentence pretraining task that can easily be combined with any encoder-based pretraining task. TSP redefines what is learned from sentence pairs via text structure relationships. Specifically, these text structure relationships combine hierarchical and ordering relationships, which can be seen as implicit learning signals for language understanding via recognition of concepts and reasoning relationships respectively. In this paper, we have shown that (1) inter-sentence baselines (NSP, SOP, SSO) were not helpful for performance at larger scales in our settings, whereas TSP improved the overall performance of models at all scales in our experiments, which again demonstrates that inter-sentence tasks for learning from inter-sentence relationships can help language understanding. Moreover, (2) the good scaling and significant improvement at the *Large* scale showed that the exploitation of text structure information can result in a sophisticated task that provides valuable knowledge for models with a large capability and data size. Finally, although there may still be much room for different ways to exploit text structure information, we have shown the effectiveness and potential of exploiting such information, and we expect that this work will encourage more research on learning from text structure information.

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

## A   PRETRAINING DETAILS

The details given in Table 3 apply to all the models in this paper. In most cases, we used the same hyperparameters as for BERT and ELECTRA. At the *Large* scale, we adopted a setting closer to BERT-Large because ELECTRA's setting for the *Large* scale requires much higher memory and computational cost than our computational resource could handle. For the *Large* scale, we also slightly increased the batch size to 288, which we found that avoids divergence in our experiments. Additionally, inspired by BERT, we trained with a sequence length of 128 for the first half of the training steps and then with a sequence length of 512 for the rest while keeping the same number of tokens in a batch to speed up pretraining in our *Large*-scale experiments. Note that we did not perform any hyperparameter search, although we think it could help our proposed method to perform better.

Table 3: Pretraining hyperparameters.

| Hyperparameter | Small | Base | Large |
|---|---|---|---|
| Number of layers | 12 | 12 | 24 |
| Hidden size | 256 | 768 | 1024 |
| FFN inner hidden size | 1024 | 3072 | 4096 |
| Attention head size | 4 | 12 | 16 |
| Attention heads | 64 | 64 | 64 |
| Embedding size | 128 | 768 | 1024 |
| Mask percent | 15 | 15 | 15 |
| Learning rate decay | Linear | Linear | Linear |
| Warmup steps | 10000 | 10000 | 10000 |
| Learning rate | 5e-4 | 2e-4 | 1e-4 |
| Adam $\epsilon$ | 1e-6 | 1e-6 | 1e-6 |
| Adam $\beta_1$ | 0.9 | 0.9 | 0.9 |
| Adam $\beta_2$ | 0.999 | 0.999 | 0.999 |
| Dropout | 0.1 | 0.1 | 0.1 |
| Weight decay | 0.01 | 0.01 | 0.01 |
| Batch size | 128 | 256 | 288 |
| Training steps (MLM/ELECTRA) | 1.45M/1M | 1M/- | 1M/- |

## B   FINETUNING DETAILS

Our finetuning setting (see Table 4 for details) mostly followed that of ELECTRA. Most tasks were finetuned for 3 epochs. However, inspired by ELECTRA, which increases the number of epochs to 10 for the RTE task, we also found that increasing the number of epochs to 10 helped stabilize the finetuning performance for CB and BoolQ. Except for this, we did not perform any hyperparameter search for finetuning.

Table 4: Finetuning hyperparameters.

| Hyperparameter | Value |
|---|---|
| Learning rate | 3e-4 for Small, 1e-4 for Base, 5e-5 for Large |
| Adam $epsilon$ | 1e-6 |
| Adam $beta\_1$ | 0.9 |
| Adam $beta\_2$ | 0.999 |
| Layerwise LR decay | 0.8 for Base/Small, 0.9 for Large |
| Learning rate decay | Linear |
| Warmup fraction | 0.1 |
| Dropout | 0.1 |
| Weight decay | 0.1 |
| Batch size | 32 |
| Training epochs | 10 for RTE, CB, BoolQ; 3 for other tasks. |

For the WiC and ReCoRD tasks, in which span embeddings for words or entities are required to compare spans with each other, we created span embeddings by applying average pooling to the final hidden representations of the tokens constructing a span and we compared spans by passing the concatenated span embeddings to the classifiers.

## C    SHUFFLING RATE

Table 5: Comparison of different sentence shuffling rates in terms of the SuperGLUE development set results for *Small*-scale MLM-based models. We choose 15% as our final choice for shuffling rate. The results show that TSP is not sensitive to the extent of shuffling, but shuffling is necessary for TSP.

| Shuffling rate | SuperGLUE |
|:---:|:---:|
| 0% | 59.7 |
| 15% | 60.3 |
| 30% | 60.3 |
| 50% | 60.2 |
| 100% | 60.1 |

Among the sentences in a given input sequence, we randomly shuffled 15% of them while other sentences stay at their original positions, to prevent complete leakage of the original ordering and hierarchical information. We also experimented with different rates for such kind of partial shuffling, and the results are listed in Table 5. The case of no shuffling and complete shuffling gave the worst results, while there was little difference in performance for shuffling of $15 \sim 50\%$. Accordingly, the results indicate that shuffling is needed, but the proposed task is not sensitive to the partial shuffling rate. We chose 15% as the partial shuffling rate in this paper because it was the lowest shuffling rate in this analysis that gave a good performance with minimal corruption of inputs.

## D    RESULTS ON THE SUPERGLUE DEVELOPMENT SET

For reference, Table 6 lists our experimental results on the SuperGLUE development set.

## E    RESULTS ON THE GLUE BENCHMARK

The GLUE benchmark is the predecessor of the SuperGLUE benchmark. Compared to the old GLUE benchmark, SuperGLUE comes with a new set of more difficult language understanding tasks and improved resources, which encourages us to evaluate models on SuperGLUE as our main results. At the same time, we also provide GLUE results for readers' reference. As shown in Table 8, the proposed task, which learns from both hierarchical and ordering relationships, also outperform other baselines on the GLUE benchmark at Small, Base, and Large scales.

## F    COMBINATION WITH OTHER WORD-LEVEL PRETRAINING METHOD

Because the proposed method can easily be added on top of other word-level pretraining methods, we conduct an experiment to evaluate whether the proposed method improves performance when it is not combined with MLM. ELECTRA (Clark et al., 2020), another word-level pretraining method, was adopted as the alternative to MLM in this experiment. We chose ELECTRA because it is a discriminative pretraining method that performs binary classification to detect replaced words among all words, making it largely different from MLM, which is generative and predicts masked words. Although ELECTRA comprises a generator to generate inputs for discrimination and a discriminator to detect replaced tokens in generated inputs, the generator is discarded after pre-training, and only the discriminator is used for finetuning. This is why, when combining TSP with ELECTRA, we applied TSP only to the discriminator and the generation stage remained unchanged.

Table 6: Results of MLM-based models on the SuperGLUE development set.

| Scale | Model | Avg. | CB | COPA | MultiRC | RTE | WIC | BoolQ | ReCoRD |
|---|---|---|---|---|---|---|---|---|---|
| | | | Avg. F1/Acc. | Acc. | F1a/EM | Acc. | Acc. | Acc. | F1/Acc. |
| Small | MLM | 58.7 | **83.9**/77.6 | 55.0 | 67.1/12.8 | 59.6 | 61.0 | 70.7 | 39.0/48.2 |
| | MLM+NSP (BERT) | 60.4 | 70.7/**80.4** | 58.0 | 67.0/14.5 | 66.4 | 65.8 | 73.0 | 38.6/48.1 |
| | MLM+SOP | 60.0 | 71.7/78.6 | 55.0 | 67.2/14.8 | 68.6 | 63.5 | 73.2 | 39.1/48.2 |
| | MLM+SSO | 60.0 | 65.0/75.0 | 56.0 | **67.3**/14.9 | 66.4 | **68.8** | 73.2 | 39.3/**49.1** |
| | MLM+TSP (ours) | **60.7** | 72.9/80.4 | **59.0** | 66.6/**15.2** | 68.6 | 62.1 | **73.5** | **39.7**/49.1 |
| Base | MLM | 71.2 | **90.6**/**92.9** | 63.0 | 75.1/27.3 | 73.7 | 66.9 | 80.1 | 64.9/78.0 |
| | MLM+NSP (BERT) | 71.6 | 89.9/**92.9** | 63.0 | 75.8/28.7 | 73.8 | 70.1 | 79.9 | 64.1/76.9 |
| | MLM+SOP | 70.2 | 83.7/87.5 | 66.0 | 75.1/26.8 | 72.6 | 68.7 | 79.1 | 62.2/74.6 |
| | MLM+SSO | 70.8 | 86.3/91.1 | 66.0 | 75.5/26.7 | 72.2 | **70.3** | 78.9 | 62.4/74.9 |
| | MLM+TSP (ours) | **73.9** | 88.1/**92.9** | **72.0** | **79.0**/**34.1** | **77.6** | 68.3 | **80.6** | **65.2**/**78.2** |
| Large | MLM | 76.4 | 93.0/94.6 | 70.0 | 80.3/35.8 | 80.7 | 70.1 | 83.3 | **71.7**/**85.5** |
| | MLM+NSP (BERT) | 76.7 | 93.7/94.6 | 70.5 | 81.5/41.6 | 81.2 | 69.7 | 82.5 | 70.7/84.2 |
| | MLM+SOP | 77.0 | 93.2/94.6 | 71.0 | 82.0/43.0 | 80.9 | 69.0 | 83.3 | 71.1/85.0 |
| | MLM+SSO | 77.3 | 93.6/94.1 | 71.5 | 82.9/43.1 | 82.0 | 70.2 | 83.3 | 70.6/84.2 |
| | MLM+TSP (ours) | **78.0** | **94.3**/**96.4** | **72.0** | **83.3**/**45.4** | 82.0 | **70.5** | **83.9** | 71.4/85.0 |

Table 7: Results of MLM-based models on the GLUE development set. "Pearson" stands for Pearson correlation. "Spearman" stands for Spearman correlation.

| Scale | Model | Avg. | CoLA | SST | MRPC | STS | QQP | MultiNLI | QNLI | RTE |
|---|---|---|---|---|---|---|---|---|---|---|
| | | | Matthew's Corr | Acc. | F1/Acc | Pearson/Spearman | F1/Acc. | matched/unmatched | Acc. | Acc. |
| Small | MLM | 77.3 | **47.0** | **90.0** | 89.4/85.0 | 83.6/83.2 | 84.6/88.5 | 79.2/80.3 | 85.2 | 59.6 |
| | MLM+NSP (BERT) | 79.1 | 44.4 | 89.8 | 90.0/85.8 | **88.4**/**88.0** | 85.4/89.1 | 79.9/81.0 | 88.3 | 66.4 |
| | MLM+SOP | 78.4 | 44.5 | 89.6 | 89.2/84.8 | 85.3/84.8 | 85.1/88.8 | 78.8/79.3 | 86.6 | 68.6 |
| | MLM+SSO | 79.2 | 45.1 | 89.6 | 90.1/85.8 | 88.2/87.8 | 85.6/89.2 | **80.3**/**81.2** | 88.7 | 66.4 |
| | MLM+TSP (ours) | **79.8** | 46.8 | 89.7 | **91.0**/**87.2** | 87.1/86.7 | 85.6/**89.3** | 80.2/**81.2** | **88.9** | 68.6 |
| Base | MLM | 84.6 | 61.5 | 93.5 | 92.2/89.1 | 89.7/89.5 | **87.9**/**91.0** | **86.6**/**86.7** | 91.8 | 73.7 |
| | MLM+NSP (BERT) | 84.3 | 60.4 | 93.1 | 91.5/88.2 | 90.4/90.1 | **87.9**/**91.0** | 85.3/85.7 | 91.7 | 73.8 |
| | MLM+SOP | 83.5 | 58.1 | 92.0 | 91.7/88.5 | 89.9/89.6 | 87.7/90.9 | 84.9/85.2 | 91.0 | 72.6 |
| | MLM+SSO | 83.9 | 60.3 | 92.5 | 91.6/88.2 | 90.7/90.3 | 87.5/90.8 | 85.0/85.5 | 91.5 | 72.2 |
| | MLM+TSP (ours) | **85.5** | **62.5** | **93.9** | **92.5**/**89.7** | 90.7/90.4 | 87.8/**91.0** | 86.3/86.6 | **92.8** | **77.6** |
| Large | MLM | 86.5 | 62.5 | 95.0 | 92.0/88.8 | 90.8/90.7 | 89.0/91.7 | **88.7**/**89.1** | 93.3 | 80.7 |
| | MLM+NSP (BERT) | 86.2 | 60.2 | 94.5 | 92.2/89.2 | 91.8/91.4 | 88.8/91.7 | 88.0/88.3 | 93.0 | 81.2 |
| | MLM+SOP | 86.5 | 61.7 | 94.6 | 92.6/89.7 | 91.6/91.4 | 88.9/91.7 | 88.3/88.6 | 93.0 | 80.9 |
| | MLM+SSO | 86.5 | 61.5 | 94.5 | 92.2/88.8 | **91.8**/**91.6** | 88.8/91.6 | 88.4/88.5 | 93.0 | **82.0** |
| | MLM+TSP (ours) | **87.4** | **64.7** | **95.1** | **93.4**/**90.9** | **91.8**/**91.6** | **89.1**/**91.9** | 88.3/89.0 | **94.1** | **82.0** |

Note that, as in the previous experiments, TSP added negligible parameters and computation to the ELECTRA pretraining.

The experimental results are listed in Table 9. When combined with ELECTRA, the proposed method could still improve the SuperGLUE score. Because of limited computational resources and time, we did not perform these experiments beyond the *Small*-scale, but we expect that the proposed method would perform well in such cases. Overall, this experiment shows the capability of the proposed method to be combined with other word-level pretraining methods, thus suggesting the opportunity to improve such methods via text structure learning in the future.

Table 8: Results of MLM-based models on the GLUE development set. "Pearson" stands for Pearson correlation. "Spearman" stands for Spearman correlation.

| Scale | Model | Avg. | CoLA Matthew's Corr | SST Acc. | MRPC F1/Acc | STS Pearson/Spearman | QQP F1/Acc. | MultiNLI matched/unmatched | QNLI Acc. | RTE Acc. |
|---|---|---|---|---|---|---|---|---|---|---|
| Small | MLM | 77.3 | **47.0** | **90.0** | 89.4/85.0 | 83.6/83.2 | 84.6/88.5 | 79.2/80.3 | 85.2 | 59.6 |
| | MLM+NSP (BERT) | 79.1 | 44.4 | 89.8 | 90.0/85.8 | **88.4/88.0** | 85.4/89.1 | 79.9/81.0 | 88.3 | 66.4 |
| | MLM+SOP | 78.4 | 44.5 | 89.6 | 89.2/84.8 | 85.3/84.8 | 85.1/88.8 | 78.8/79.3 | 86.6 | 68.6 |
| | MLM+SSO | 79.2 | 45.1 | 89.6 | 90.1/85.8 | 88.2/87.8 | 85.6/89.2 | **80.3/81.2** | 88.7 | 66.4 |
| | MLM+TSP (ours) | **79.8** | 46.8 | 89.7 | **91.0/87.2** | 87.1/86.7 | 85.6/89.3 | 80.2/81.2 | **88.9** | 68.6 |
| Base | MLM | 84.6 | 61.5 | 93.5 | 92.2/89.1 | 89.7/89.5 | 87.9/91.0 | 86.6/86.7 | 91.8 | 73.7 |
| | MLM+NSP (BERT) | 84.3 | 60.4 | 93.1 | 91.5/88.2 | 90.4/90.1 | 87.9/91.0 | 85.3/85.7 | 91.7 | 73.8 |
| | MLM+SOP | 83.5 | 58.1 | 92.0 | 91.7/88.5 | 89.9/89.6 | 87.7/90.9 | 84.9/85.2 | 91.0 | 72.6 |
| | MLM+SSO | 83.9 | 60.3 | 92.5 | 91.6/88.2 | 90.7/90.3 | 87.5/90.8 | 85.0/85.5 | 91.5 | 72.2 |
| | MLM+TSP (ours) | **85.5** | **62.5** | **93.9** | **92.5/89.7** | **90.7/90.4** | 87.8/91.0 | 86.3/86.6 | **92.8** | **77.6** |
| Large | MLM | 86.5 | 62.5 | 95.0 | 92.0/88.8 | 90.8/90.7 | 89.0/91.7 | **88.7/89.1** | 93.3 | 80.7 |
| | MLM+NSP (BERT) | 86.2 | 60.2 | 94.5 | 92.2/89.2 | 91.8/91.4 | 88.8/91.7 | 88.0/88.3 | 93.0 | 81.2 |
| | MLM+SOP | 86.5 | 61.7 | 94.6 | 92.6/89.7 | 91.6/91.4 | 88.9/91.7 | 88.3/88.6 | 93.0 | 80.9 |
| | MLM+SSO | 86.5 | 61.5 | 94.5 | 92.2/88.8 | **91.8/91.6** | 88.8/91.6 | 88.4/88.5 | 93.0 | **82.0** |
| | MLM+TSP (ours) | **87.4** | **64.7** | **95.1** | **93.4/90.9** | 91.8/91.6 | **89.1/91.9** | 88.3/89.0 | **94.1** | 82.0 |

Table 9: Results of ELECTRA-based models on the SuperGLUE development set. TSP performs well even when combined with another word-level pretraining method that is vastly different from MLM.

| Scale | Model | SuperGLUE |
|---|---|---|
| Small | ELECTRA | 57.4 |
| | ELECTRA+NSP | 58.6 |
| | ELECTRA+SOP | 58.5 |
| | ELECTRA+SSO | 60.6 |
| | ELECTRA+TSP | **60.7** |

