# OpenReview forum: "Improving Language Model Pretraining with Text Structure Information"
_ICLR.cc/2023/Conference — Submitted to ICLR 2023_

### Official Review · Reviewer_SQUJ · 2022-10-20

**Confidence:** 4
**Correctness:** 2
**Technical Novelty And Significance:** 2
**Empirical Novelty And Significance:** 2
**Recommendation:** 3

**Clarity, Quality, Novelty And Reproducibility:**

* Clarity: The paper is clearly written and well-organized.
* Quality: The contribution of the paper is not convincing due to the problematic empirical evaluation (see cons above).
* Novelty: The novelty is quite limited. The proposed objective TSP is essentially a combined version of previous sentence-level tasks (see cons above).


**Strength And Weaknesses:**

Pros:
* The paper is clearly written and well-organized.
* The proposed pretraining task TSP is shown to outperform the compared sentence-level pretraining objectives (NSP, SOP and SSO) on SuperGLUE.

Cons:
* The novelty of the method is quite limited. The proposed objective TSP is essentially a combined (and slightly extended) version of NSP and SOP -- the "hierarchical relation" classification is extending NSP by further classifying the "non-neighbor" case into "same-paragraph" and "same-document"; the "order relation" classification is equivalent to SOP. In this sense, the proposed task is not new and should be compared with multi-task training that uses both NSP and SOP (this seems to be an important baseline that is missing from comparisons).
* The paper omitted discussions and comparisons with an important family of sentence-level pretraining tasks that use contrastive learning. For example, COCO-LM and CLEAR demonstrated that contrastive learning-based sequence-level tasks can effectively improve NLU task fine-tuning performance, and they can also be combined with word-level pretraining tasks.
* The empirical evaluation is quite problematic. My major concern comes from the fact that the paper does not follow the conventional pretraining/downstream task settings that have been extensively used by previous studies. (1) The pretraining corpus used in this paper is OpenWebText whereas previous pretraining methods use either Wikipedia + BookCorpus (in BERT) or further expanded corpora that include a wide variety of datasets besides OpenWebText (in RoBERTa). The common pretraining corpora choice could be either of the two above. Even if BookCorpus is not publicly available, Wikipedia should at least be included as a formal language pretraining source. If the pretraining experiments are conducted only on OpenWebText, it's very unclear whether the observed performance gain of TSP can be generalized to common/larger pretraining corpora. (2) Most pretrained language models (e.g., BERT, RoBERTa, ELECTRA) have been evaluated on GLUE whereas this paper only reports SuperGLUE results. This makes it hard to compare with previous pretrained models. While it is good to have SuperGLUE results, I believe it's important to conduct experiments on GLUE as well to facilitate easy comparisons with prior work. (3) The evaluation results only have the best performing results on the test set but not median results across all tasks (Table 6 only has averaged scores). This is problematic because some tasks have very small test sets and the scores could vary a lot. The common practice is to report median results over several random seeds on dev sets of all tasks.

References:
Meng, Yu et al. “COCO-LM: Correcting and Contrasting Text Sequences for Language Model Pretraining.” NeurIPS (2021).
Wu, Zhuofeng et al. “CLEAR: Contrastive Learning for Sentence Representation.” ArXiv abs/2012.15466 (2020).

---
**Post-Rebuttal Updates**:
I thank the authors for providing responses to my raised concerns. Unfortunately, I'm not convinced by the authors' arguments regarding my major concerns. My detailed comments are as below:
* Regarding novelty: I never implied in my review that a method needs to be complicated to be novel; instead, simple methods can be novel as long as they provide new perspectives/insights. My critique regarding the novelty of the paper is mainly due to the fact that the proposed objective TSP is very similar to previously proposed sentence-level objectives (NSP and SOP) in formulation. The major difference between TSP and NSP+SOP, according to the authors' response, is that TSP considers paragraph-level relationships. However, the motivation for this idea is very vague -- how can one accurately/unambiguously define the real difference between a sentence and a paragraph, and the difference between a paragraph and a document? For example, in novels, it's very common that one or two sentences can be a paragraph; other times, a paragraph can also be an entire document. If the distinction between sentences, paragraphs and documents is even hard to tell for humans, then I don't see how the so-called "paragraph-level relationships" newly introduced in TSP can actually make a big difference compared to a simple combination of previous methods (NSP+SOP).
* Regarding the comparisons to contrastive learning approaches: I'm not fully convinced by the argument that the comparisons between contrastive learning objectives and the proposed TSP objective can be omitted. I understand that the two sets of methods are based on different principles, but they are indeed for the same goal -- improving the sequence-level representations of PLMs. Therefore, unless TSP has other usages/benefits that contrastive learning fails to bring, I don't see why they cannot be fairly compared to each other. Such comparisons do not exist in prior work probably because it is quite well known that NSP/SOP-style methods do not really work well (also shown in this paper), and many PLMs simply ended up not using any sequence-level objectives (e.g., RoBERTa, ELECTRA), so the contrastive learning papers didn't compare with NSP/SOP (of course, I do hope to see those results in the contrastive learning papers).
* The empirical gain of TSP does not seem convincing to me: Although TSP provides significant gains on GLUE/SuperGLUE average score, the gains are mainly from small/unstable tasks, like CoLA, MRPC, RTE and STS-B. Fine-tuning on those tasks are notoriously unstable and is subject to high variance. On large tasks like MNLI and SST-2 (note that they are more reliable than the previous tasks; RoBERTa paper uses these two tasks intead of the average GLUE score for ablation studies), the TSP objective barely provides any gain and even worsens the result. Therefore, I suspect that the performance gain of TSP is simply due to randomness in fine-tuning.
* Regarding the choice of pretraining corpus (OpenWebText): I'm indeed aware that OpenWebText is larger than Wikipedia, but Wikipedia is still a more standard choice for pretraining corpus if only one corpus is used (e.g., base model exploration in BERT, RoBERTa, ELECTRA all used Wikipedia + BookCorpus). This is probably because Wikipedia's text quality is arguably higher than OpenWebText which contains many short/casual/informal texts. Therefore, I'd suggest the authors follow the standard protocol for pretraining; otherwise, the findings and conclusions may not be reliable generalize well. This is not my major concern about the paper though.


**Summary Of The Paper:**

The paper proposes an inter-sentence pretraining task, Text Structure Prediction (TSP), which encourages the encoder to classify sentence structure relationships. The relationships are manually defined to be among 6 types across two dimensions: hierarchical relation and ordering relation. The hierarchical relation contains three cases: neighbor, same paragraph and same document; the ordering relation contains two cases: original and reversed order. By randomly shuffling a subset of the sentences in a sequence, TSP can be used to train the encoder to classify sentence pairs into the pre-defined 6 relation types. The authors argue that such an inter-sentence pretraining task provides more high-level language understanding signals and thus improves using word-level pretraining tasks only. The major benefit of TSP compared to previously proposed inter-sentence pretraining task, as pointed out by the authors, is that TSP is able to offer performance gain for encoders of different sizes (small, base, large), whereas previous methods usually do not benefit large model sizes. The empirical evaluation is conducted on the SuperGLUE benchmark. The authors show that TSP is more effective than other compared sentence-level tasks when combined with MLM.

**Summary Of The Review:**

I believe the paper will need quite a lot major revisions to be convincing. (1) There should be more discussions and evaluations regarding what are the real differences from NSP and SOP. (2) The method needs to be compared with contrastive-learning based pretraining tasks. (3) The evaluation needs to follow conventional pretraining/fine-tuning settings.

---

> ### Author Response · Authors · 2022-11-13
> **Response to reviewer SQUJ**
>
> Thank you for the comments. We address some of the concerns and questions below. Due to the length of our reply, we divided it into multiple comments as follows:

---

> > ### Author Response · Authors · 2022-11-13
> > **Response to reviewer SQUJ (part 1)**
> >
> > > _“The novelty of the method is quite limited.”_
> >
> > **Response 1: Our novelty focuses on providing a new viewpoint and promising research direction, while the proposed method is designed to be simple but non-trivial for showcasing the usefulness and potential of our viewpoint:** \
> >
> > As stated in the introduction, contribution, and conclusion sections, our goal is not to achieve SOTA by proposing a complex method. Instead, we want to show the utilization of text structure as learning signals is a promising research direction, and the proposed method is designed to demonstrate it.
> >
> > &nbsp;
> >
> > > “The proposed objective TSP is essentially a combined (and slightly extended) version of NSP and SOP -- the ‘hierarchical relation’ classification is extending NSP by further classifying the "non-neighbor" case into "same-paragraph" and ‘same-document’; the ‘order relation’ classification is equivalent to SOP. ”_
> >
> > **Response 2: Our method captures a complete picture of text structure and can not be described by the addition of previous works:** \
> > Unlike a simple addition of NSP and SOP without the concept of text structure in mind,  the proposed framework is different from it in two perspectives. Firstly, unlike NSP+SOP, which thinks of order and hierarchy independently, the framework includes the interaction of order and hierarchy, which captures the complete picture of text structure. For example, the order of paragraphs, which can represent high-level relationships between texts as shown in Figure 2, is not addressed in either NSP or SOP but in our framework. Secondly, NSP and SOP split the given sequence into two randomly split big sentences, whereas we create many sentences from a sequence by punctuation-based rules (using NLTK) as described in section 3.1.1, which can be more natural for learning from text structure. As we described in the following response, the addition of NSP and SOP is actually SSO and it is not competitive with our method in the experiments.
> >
> > &nbsp;
> >
> > > _”In this sense, the proposed task is not new and should be compared with multi-task training that uses both NSP and SOP (this seems to be an important baseline that is missing from comparisons).”_
> >
> >
> > **Response 3: SSO, which uses both NSP and SOP, is already included in our baselines:** \
> > Sentence Structure Objective (SSO) [1] learns from distinguishing from the original text, sentences from other documents, and sentences in reversed order. Our experiment results have shown that SSO is not competitive with our method, which demonstrates that our method can not be summarized as an addition of NSP and SOP.
> >
> > &nbsp;
> >
> > > _“The paper omitted discussions and comparisons with an important family of sentence-level pretraining tasks that use contrastive learning. For example, COCO-LM and CLEAR demonstrated that contrastive learning-based sequence-level tasks can effectively improve NLU task fine-tuning performance, and they can also be combined with word-level pretraining tasks.”_
> >
> > **Response 4: We have strengthened the emphasis on the existing discussion about contrastive learning methods:** \
> > Thank you for pointing out the importance of discussion of contrastive learning methods. We have already had an introduction about different kinds of sentence-level pretraining, including contrastive learning, in 2.2 SENTENCE-LEVEL PRETRAINING. We have modified the section to emphasize the categorization and discuss more on how each approach work. We would also like to appreciate your mention of COCO-LM [4] and CLEAR [5]. We decided to cite them as they are good examples of contrastive sentence-level pretraining tasks.
> >
> >
> > **Response 5: It is not a common practice to compare approaches that belong to different categories, benefit from different principles, and are possibly not mutually exclusive.** \
> > For comparison, there is currently no research comparing the three kinds of methods. For example, you can not see CMLM [7] (a sentence bottleneck approach), COCO-LM [4] and CLEAR [5] (contrastive learning approaches) compare themselves to inter-sentence approaches like NSP, SOP, or SSO. On the contrary, we did not see SOP (proposed in the ALBERT paper) and SSO (proposed in the StructBERT paper) compare themselves to any contrastive learning approaches. We think that is because the three approaches have their own principles and benefits differently. For example, contrastive learning benefits from the ease of the anisotropy problem while the inter-sentence approach is the only approach to encode and contextualize multiple sentences. Moreover, these different directions of sentence-level learning may not be mutually exclusive. To be honest, designing a task or framework that unifies and get all the different benefits of the approaches is one of my future research direction.

---

> > > ### Author Response · Authors · 2022-11-13
> > > **Response to reviewer SQUJ (part 2)**
> > >
> > > > _“The pretraining corpus used in this paper is OpenWebText whereas previous pretraining methods use either Wikipedia + BookCorpus (in BERT) or further expanded corpora that include a wide variety of datasets besides OpenWebText (in RoBERTa). The common pretraining corpora choice could be either of the two above. Even if BookCorpus is not publicly available, Wikipedia should at least be included as a formal language pretraining source. If the pretraining experiments are conducted only on OpenWebText, it's very unclear whether the observed performance gain of TSP can be generalized to common/larger pretraining corpora.”_
> > >
> > > **Response 6-1: OpenWebText is larger than English Wikipedia:** \
> > > OpenWebText has 38~40Gb raw texts¹ while English Wikipedia has 33GB raw texts². For another reference, considering generated dataset sizes (data stored in .arrow format) that huggingface/datasets³ reported, OpenWebText has a size of  39.7GB while Wikipedia-en and bookcorpus have a size of 20GB and 4GB respectively.
> > >
> > > **Response 6-2: OpenWebText is used as pretrainig data in many other works:** \
> > > OpenWebText is a replication of WebText dataset, which is proposed and used by GPT2 [8]. OpenWebText is also used by ELECTRA [9] (the results are published on the official github site⁴), PMI-Masking [10], SCRIPT [11], COCO-LM [4], etc.
> > >
> > > &nbsp;
> > >
> > > >_” Most pretrained language models (e.g., BERT, RoBERTa, ELECTRA) have been evaluated on GLUE whereas this paper only reports SuperGLUE results. This makes it hard to compare with previous pretrained models. While it is good to have SuperGLUE results, I believe it's important to conduct experiments on GLUE as well to facilitate easy comparisons with prior work.”_
> > >
> > > **Response 7-1: GLUE results are added:** \
> > >  We added GLUE results as Appendix E, where we can find the proposed method doing well also on the GLUE benchmark.
> > >
> > > **Response 7-2: Matters needing attention for comparing GLUE results:** \
> > > We would like to softly remind readers that when comparing GLUE results across different papers, you will want to make sure the computation cost and the number of parameters for pretraining are the same. For finetuning, whether hyper-parameter search, intermediate-finetuning, or special tricks are performed, how many finetuning runs for each task, and additional data are added, all of these are tricky points of comparing GLUE results.
> > >
> > > &nbsp;
> > >
> > > > _” The evaluation results only have the best performing results on the test set but not median results across all tasks (Table 6 only has averaged scores). This is problematic because some tasks have very small test sets and the scores could vary a lot. The common practice is to report median results over several random seeds on dev sets of all tasks.”_
> > >
> > > **Response 8: We have added development set results that report median scores for each task:** \
> > > We have expanded table 6 to report median scores on the development sets for each task. We have observed a roughly similar pattern to our main results.
> > >
> > > &nbsp;
> > >
> > > References \
> > > [1] Wang et al., “StructBERT: Incorporating Language Structures into Pre-training for Deep Language Understanding.” ICLR 2020. \
> > > [2] Devlin et al., “BERT: Pre-training of Deep Bidirectional Transformers for Language Understanding.” NAACL 2019. \
> > > [3] Lan et al., “Albert: A lite bert for self-supervised learning of language representations.” ICLR 2020. \
> > > [4] Meng et al., “COCO-LM: Correcting and Contrasting Text Sequences for Language Model Pretraining.” NeurIPS 2021. \
> > > [5] Wu et al., “CLEAR: Contrastive Learning for Sentence Representation.” arXiv 2020. \
> > > [6] Gao et al., “SimCSE: Simple Contrastive Learning of Sentence Embeddings.” EMNLP 2021. \
> > > [7] Yang et al., “Universal Sentence Representation Learning with Conditional Masked Language Model.” EMNLP 2021. \
> > > [8] Radford et al.“Language Models are Unsupervised Multitask Learners”, 2019. \
> > > [9] Clark et al., “ELECTRA: Pre-training Text Encoders as Discriminators Rather Than Generators” ICLR 2020. \
> > > [10] Levine et al., “PMI-Masking: Principled masking of correlated spans.” ICLR 2021. \
> > > [11] Nijkamp et al., “SCRIPT: Self-Critic PreTraining of Transformers” NAACL 2021. \
> > >
> > > Footnotes \
> > > ¹ https://skylion007.github.io/OpenWebTextCorpus/  \
> > > ² https://en.wikipedia.org/wiki/Wikipedia:Size_in_volumes \
> > > ³ https://huggingface.co/docs/datasets \
> > > ⁴ https://github.com/google-research/electra

---

> ### Comment · Reviewer_SQUJ · 2022-12-12
> **Post-Rebuttal Updates**
>
> I'd like to thank the authors for their responses. Unfortunately, my major concerns remain (please see **Post-Rebuttal Updates** in my main review above). I hope the authors could consider the raised points in the next major revision of the paper.

---

### Official Review · Reviewer_t9DH · 2022-10-23

**Confidence:** 4
**Correctness:** 3
**Technical Novelty And Significance:** 2
**Empirical Novelty And Significance:** 2
**Recommendation:** 5

**Clarity, Quality, Novelty And Reproducibility:**

The clarity is good. The quality and novelty are fair. The experiments seem reproducible.

**Strength And Weaknesses:**

Strength:
1. The paper is clearly stated and easy to follow.

2. The performance improves significantly in some of the SuperGLUE downstream tasks.

Weaknesses:
1. The hierarchical relationship labels are not convincing enough. It is even difficult for humans to predict whether two sentences are from the same document. Enforcing the model to learn the "same document" label might cause the overfitting of the sentence representations. The ablation study also shows that order information is more important than hierarchical structure information.

2. The empirical results are not uniformly improved with the proposed TSP methods on the downstream tasks.

**Summary Of The Paper:**

The paper proposed a new inter-sentence training task for language model pertaining, called textual structure prediction. The author designs six structure relationship labels for TSP prediction. The empirical results show that the TSP improves the performance of pretrained language models on NLU tasks.

**Summary Of The Review:**

The paper proposed hierarchical structure labels for training language models. The empirical results show performance improvements, while why the hierarchical labels work needs more reasonable explanations.

---

> ### Author Response · Authors · 2022-11-13
> **Response to reviewer t9DH**
>
> Thank you for your thoughtful review. Due to the length of our reply, we divided it into multiple comments as follows:

---

> > ### Author Response · Authors · 2022-11-13
> > **Response to reviewer t9DH (part1)**
> >
> > > _“The hierarchical relationship labels are not convincing enough. It is even difficult for humans to predict whether two sentences are from the same document. Enforcing the model to learn the ‘same document’ label might cause the overfitting of the sentence representations. The ablation study also shows that order information is more important than hierarchical structure information.”_
> >
> > **Response 1: We have made the description clear that the task is meaningfully solvable through contextualized sentence representations:** \
> > Thank you for pointing out parts that are not well explained. Notice that we do not create sentence representation for each sentence independently but encode a group of 10~20 sentences (the number depends on sentence lengths and the max sequence length) together to get the sentence representation for each sentence, as we try to emphasize in section 3.1.2 SENTENCE EMBEDDING. This means when we compare two sentence representations for their hierarchical relationships, not only the two sentences themselves but also their contexts encoded in the representations are in consideration. Intuitively, discriminating hierarchical and ordering relationships are hard if considering only two sentences, we thus expect that this motivates models to learn complex interactions between sentences in the context to encode useful context information to solve the task. We appreciate you for pointing out where other readers may also be unclear. To make it clear, we have added a description in the aforementioned section to explicitly explain the role of contextualized sentence representation in solving the proposed task.
> >
> > **Response 2: It could be arguable that identifying whether two sentences come from the same document is hard:** \
> > The authors in the ALBERT paper (Lan et al., 2020) state identifying whether two sentences come from the same document is too simple, which becomes the motivation for the authors to propose the SOP task.
> >
> > **Response 3: Overfitting may not fit in this case:** \
> > As far as I know, overfitting means that models are doing too well on the trainset and lose the generalization ability to unseen data. One of its common reason is the lack of training data. Since you suggest models may perform badly on the proposed task and the pretraining is generally considered with large training data. Overfitting may not fit in this case.
> >
> > **Response 4: Results showed that learning hierarchical relationships improve performance, and it could have large potential when training on more structured documents:** \
> > Notice that the ablation study has shown that learning hierarchical relationships gives a notable rise in performance. Besides, we have found that our pretraining data OpenWebText is mostly composed of news, which often has short paragraphs and is not as structured as other documents like Wikipedia posts. Although we need more experiments, it is intuitive that learning hierarchical relationships can contribute more when training on more structured data. But back to our original motivation, it is clear from the existing results that learning hierarchical relationships improves performance, which opens a promising research direction for further improvement either by curated data or more sophisticated task design.

---

> > > ### Author Response · Authors · 2022-11-13
> > > **Response to reviewer t9DH (part 2)**
> > >
> > > > _“The empirical results are not uniformly improved with the proposed TSP methods on the downstream tasks.”_
> > >
> > > **Response 5: It could be surprisingly hard and not common that GLUE/SuperGLUE tasks are improved uniformly:** \
> > > We would like to softly remind readers that improving on every task or improving largely on every task in the GLUE / SuperGLUE benchmarks is hard and not common in language model pretraining papers. Examples are as follows: In Table 2 of the ELECTRA paper [1], ELECTRA did not significantly outperform BERT on the SST task under the same computation cost. In Table 8 of the PMI-Masking paper [2], PMI-Masking did not outperform its baselines on 4 of 8 tasks. In Tables 1 and 5 of the DeBERTa paper [3], DeBERTa was unable to outperform its counterparts on 2 GLUE tasks and 3 SuperGLUE tasks. Besides, in our experiments, the proposed method is the only one that improves or is competitive to pure MLM on almost all tasks at Small, Base, and Large scales.
> > >
> > > &nbsp;
> > >
> > > > _“The empirical results show performance improvements, while why the hierarchical labels work needs more reasonable explanations.”_
> > >
> > > **Response 6: The explanation of hierarchical relationship learning has been renewed.** \
> > > Thank you for pointing out parts that are not well explained. We have rewritten explanations of why learning hierarchical relationships works in section 3.3 to make it clearer. To group sentences nestedly, models are expected to summarize semantics at different levels from the context, while capturing relationships and interactions between these semantics. Intuitively, it can provide unique learning signals for models' understanding ability on and between texts at different levels of granularity. See the complete explanation in renewed section 3.3.
> > >
> > > &nbsp;
> > >
> > > Reference: \
> > > [1] Clark et al., “ELECTRA: Pre-training Text Encoders as Discriminators Rather Than Generators.” ICLR 2020. \
> > > [2] Levine et al., “PMI-Masking: Principled Masking of Correlated Spans.” ICLR 2021. \
> > > [3] He et al., “DeBERTa: Decoding-enhanced BERT with Disentangled Attention.” arXiv 2020.

---

### Official Review · Reviewer_LMAd · 2022-10-24

**Confidence:** 4
**Correctness:** 3
**Technical Novelty And Significance:** 3
**Empirical Novelty And Significance:** 2
**Recommendation:** 8

**Clarity, Quality, Novelty And Reproducibility:**

* Well-written and easy to follow.
* The TSP task is not hugely different from prior sentece-based pretraining methods, but it does work.
* See the point in Weaknesses and the Punkt sentece splitter.
* The paper contains all the hyperparameters needed to reproduce the experiments.


**Strength And Weaknesses:**

## Strengths

* Well-written and easy to follow.
* The models are properly trained before evaluation. Section 4.1.1 states, "To ensure fair comparisons, all models at the same scale had almost the same computational cost and number of parameters in both training and inference.". This is a detail people often forget, so I am happy to see that the paper takes computation into account.

## Weaknesses

* The introduction argues that "Without text structure, a text becomes a long continuous word sequence, which is hard to read and makes it difficult to identify key concepts and logical relationships." While this is probably, intuitively, true for humans, the paper uses the statement to argue against prior unsupervised sentence tasks and projects intuitive understanding of human attention and reading abilities onto machines (i.e., language models) without valid proof. This is a logical fallacy. I do not believe that this is a major weakness for the paper. The paragraph just needs to be rewritten to avoid the incorrect argument.
* The Related Work / Word-Level Pretraining section is a little too naroow. I understand that the paper's method only applies to transformer encoders, but I believe the paper would benefit from summarizing pretraining methods for encoder-decoders too.
* Section 4.1.1 cites "the sentence splitter of NLTK (Bird et al., 2009) for the use of TSP". NLTK's sentence splitter is Punkt (Kiss and Strunk, 2006), see also https://www.nltk.org/_modules/nltk/tokenize/punkt.html. The paper should cite the Punkt paper together with the NLTK citation. Giving credit to the method's authors is not only fair, but it explicitly names the sentence splitter used, thus improving clarity for readers.


**Summary Of The Paper:**

The paper argues for adding inter-sentence tasks to language model pretraining. In this direction, the paper introduces Text Structure Prediction (TSP), an improved version of BERT's Next Sentence Prediction where a LM must classify two sentences using one of six labels that track whether sentences belong to the same document, same paragraph, are neighbors and their order (A then B vs. B then A).

**Summary Of The Review:**

A good paper with a couple of minor issues that can be easily addressed before a camera ready.

---

> ### Author Response · Authors · 2022-11-13
> **Response to reviewer LMAd**
>
> Thank you for your encouraging feedback!
>
> &nbsp;
>
> > _“The introduction argues that ‘Without text structure, a text becomes a long continuous word sequence, which is hard to read and makes it difficult to identify key concepts and logical relationships.’ While this is probably, intuitively, true for humans, the paper uses the statement to argue against prior unsupervised sentence tasks and projects intuitive understanding of human attention and reading abilities onto machines (i.e., language models) without valid proof. This is a logical fallacy. I do not believe that this is a major weakness for the paper. The paragraph just needs to be rewritten to avoid the incorrect argument.”_
>
> **Response 1: The argument was rewritten to improve its preciseness:** \
> Thank you for pointing it out. We agree that the current statement can be misleading. We have rewritten the related paragraph as follows: “Without text structure, a text becomes a long continuous word sequence, which is hard to read and makes it difficult to identify key concepts and logical relationships for humans. This intuitive understanding of human reading ability inspires us to explore the possibility that text structure can provide models' language understanding ability abundant learning signals of high-level semantics and their interaction, especially for models at larger scales.”
>
> &nbsp;
>
> > _“The Related Work / Word-Level Pretraining section is a little too narrow. I understand that the paper's method only applies to transformer encoders, but I believe the paper would benefit from summarizing pretraining methods for encoder-decoders too.”_
>
> **Response 2: A short summary of encoder-decoder-based pretraining methods has been added:** \
> Thank you for your suggestion. We added a short summary of pretraining methods that are based on transformer encoder-decoder in the mentioned section. This should help readers better understand the architectural aspect of pretraining methods.
>
> &nbsp;
>
> > _“Section 4.1.1 cites ‘the sentence splitter of NLTK (Bird et al., 2009) for the use of TSP’. NLTK's sentence splitter is Punkt (Kiss and Strunk, 2006), see also https://www.nltk.org/_modules/nltk/tokenize/punkt.html. The paper should cite the Punkt paper together with the NLTK citation. Giving credit to the method's authors is not only fair, but it explicitly names the sentence splitter used, thus improving clarity for readers.”_
>
> **Response 3: The citation of Punkt has been added:** \
> Thank you for pointing out our miss. We deeply believe that not only the distribution but also the original work that it is based on should be cited. Also, we found citing Punkt should let readers understand how sentences are split and the result of splitting better. In the section Proposed Method/Proposed Framework/Input sequence, we added a citation for Punct and make it clear that NLTK sentence splitter is based on Punkt.

---

### Official Review · Reviewer_VZCD · 2022-10-25

**Confidence:** 4
**Correctness:** 4
**Technical Novelty And Significance:** 3
**Empirical Novelty And Significance:** 3
**Recommendation:** 6

**Clarity, Quality, Novelty And Reproducibility:**

The proposed method is clearly described. The novelty is enough for me. They propose additional self-supervised loss to improve the language models.

**Strength And Weaknesses:**

Strength
- The motivation is clear and reasonable.
- The experiments support the claim and seem to be promising

Weaknesses
- To test the generalizability, I suggest to test on GLUE tasks as well.
- It's not clear what size of models is using for the ablation study in Table 2
- What will happen if we do not shuffle the sentences?

**Summary Of The Paper:**

This paper proposes a new way to pre-train language models. The main idea is to include sentence-level hierarchy information during the pre-training. Instead of considering only neighbor sentences, they consider more relations between sentences such as if they are in the same paragraph or if they are in the same document. By training this classification loss along with the original masked token loss, they have better performance on the SuperGLUE downstream tasks.

**Summary Of The Review:**

The motivation is clear and reasonable. Although the proposed method is simple, it's very effective. Ablation study supports their claim.

---

> ### Author Response · Authors · 2022-11-13
> **Response to reviewer VZCD**
>
> Thank you for your supportive feedback.
>
> > _“To test the generalizability, I suggest to test on GLUE tasks as well.”_
>
> **Response 1: The GLUE results were added:** \
> Thank you for your suggestion. We have added evaluation results on the GLUE benchmark in Appendix E. The proposed task outperforms the baselines also on the GLUE benchmark, which showed that the usefulness of learning text structure information is not restricted to the SuperGLUE tasks.
>
> > _“It's not clear what size of models is used for the ablation study in Table 2”_
>
> **Response 2: The description now mentions “Base scale”:** \
> Thank you for pointing that out. We modified the description referring to Table 2. The description is now: "The results on the SuperGLUE development set of models at the Base scale are shown in Table2.
>
> > _“What will happen if we do not shuffle the sentences?”_
>
> **Response 3: Not shuffling the sentences hurts performance in our experiment results:** \
> Thank you for raising an interesting question! As we have pointed out in Appendix C, not shuffling the sentences (0% shuffling rate) gives the worst performance in models trained with different shuffling rates. This indicates that adding noise to the text structure information in the inputs is essential for learning to predict the text structures.

---

> > ### Comment · Reviewer_VZCD · 2022-11-17
> > **Thanks for the response.**
> >
> > Thanks for your response. It addresses my questions.

---

### Author Response · Authors · 2022-11-13
**Paper Updates**

We would like to thank all the reviewers again for their thoughtful reviews and constructive feedback, which have undoubtedly improved the quality of our paper. The reviewers agreed the work is well-motivated and well-written (vzcd, lmad, t9dh, squj), and has good performance that is promising and supports the claims (vzcd, lmad, t9dh, squj) under fair and reproducible setting (lmad, t9dh). On the other hand, we have added new results and revisions to address reviewers’ concerns as follows:
- We have added results on GLUE. The results showed the proposed method perform also well on GLUE. Additionally, we have added development results for each task in the benchmarks.
- We have fixed the missing mention of model size in Table 2, and the missing citation of Punkt.
Revise the argument on the analogy of human reading and machine learning to avoid misleading understanding as reviewer lmad has pointed out.
- We have made the related work disscusion on word-level pretraining more informative. We included an architectural discussion including encoder-decoder in section 2.1.
- We have made the related work disscusion on sentence-level pretraining more informative. We included a deeper discussion on different categories of sentence-level pretraining including contrastive learning in section 2.2.
- We explained why the task is meaningfully solvable by emphasizing more on the contextualization of sentence representation and its role in identifying hierarchical relationships in Section 3.1.2.
- We have made it clearer why learning hierarchical relationships can help. We rewrote the meaning of learning hierarchical relationships in Section 3.3.

---

### Decision · Program_Chairs · 2023-01-20

**Decision:**

Reject

**Justification For Why Not Higher Score:**

The paper doesn't really compare with previously reported results. They execution could be different and they could have picked a standard pretraining corpus like Books+Wiki and then directly compare with prior works. In addition, some of the very closely related work is missing both in discussion and comparison (https://arxiv.org/pdf/2205.01703.pdf). Some of the reviewers remain unconvinced after the author response.

**Justification For Why Not Lower Score:**

N/A

**Metareview: Summary, Strengths And Weaknesses:**

The paper presents a new approach to pre-training language models, incorporating sentence-level hierarchy information during the pre-training process. The proposed method, called Text Structure Prediction (TSP), involves classifying sentences based on their relation to one another, such as whether they are in the same paragraph or document. The results of the experiments indicate that the TSP method leads to improved performance compared with other pretraining methods.

Overall, the paper has several strengths, including a clear motivation for the proposed approach and well-supported claims through experiments. However, some weaknesses have also been identified by the reviewers, including a lack of proper empirical evaluation and choice of a different pretraining dataset than those used in prior models, which doesn't allow for direct comparison with previously reported results. Additionally, the paper could benefit from a more comprehensive discussion and comparison with close related work (e.g., contrastive learning papers and other structured pretraining methods like SLM; Lee et al, 2020).